# Use of Botulinum Toxin in Upper-Limb Tremor: Systematic Review and Perspectives

**DOI:** 10.3390/toxins16090392

**Published:** 2024-09-13

**Authors:** Damien Motavasseli, Cécile Delorme, Nicolas Bayle, Jean-Michel Gracies, Emmanuel Roze, Marjolaine Baude

**Affiliations:** 1Assistance Publique Hôpitaux de Paris, Service de Rééducation Neurolocomotrice, Hôpitaux Universitaires Henri Mondor, F-94010 Créteil, France; 2UR 7377 BIOTN, Laboratoire Analyse et Restauration du Mouvement, Université Paris Est Créteil (UPEC), F-94010 Créteil, France; 3Assistance Publique Hôpitaux de Paris, Institut de Neurologie, Groupe Hospitalier Pitié-Salpêtrière, F-75013 Paris, France; 4Paris Brain Institute, INSERM, CNRS, Sorbonne University, F-75013 Paris, France

**Keywords:** tremor, botulinum toxin, ataxia, cerebellar, essential tremor, dystonic tremor, resting tremor, action tremor

## Abstract

Background: Tremor is the most common movement disorder, with significant functional and psychosocial consequences. Oral medications have been disappointing or limited by side effects. Surgical techniques are effective but associated with risks and adverse events. Botulinum toxin (BT) represents a promising avenue but there is still no double-blind evidence of efficacy on upper limb function. A systematic review on the effects of BT in upper-limb tremor was conducted. Methods: A systematic search of the literature was conducted up to July 2023, including the keywords “botulinum toxin” and “tremor”. All randomized controlled trials (RCTs) and open-label studies were analyzed. Independent reviewers assessed their methodological quality. Results: There were only eight published RCTs and seven published open-label studies, with relatively small sample sizes. This review suggests that BT is more effective when injections are patient-tailored, with analyses based on clinical judgement or kinematics. Subjective and objective measures frequently improve but transient weakness may occur after injections, especially if wrist or fingers extensors are targeted. A number of studies had methodological limitations. Conclusions: The authors discuss how to optimize tremor assessments and effects of BT injection. Controlled evidence is still lacking but it is suggested that distal “asymmetric” BT injections (targeting flexors/pronators while sparing extensors/supinators) and proximal injections, involving shoulder rotators when indicated, may avoid excessive weakness while optimizing functional benefit.

## 1. Introduction

Tremor, defined as an involuntary and rhythmic oscillatory movement of any body part, is the most common movement disorder in adults, and typically predominates in upper limbs [1]. Tremors are classified into two main categories [2,3]. The first is resting tremor, which is not voluntarily activated and is assessed when the affected part of the body is relaxed, if possible with full support against gravity. The second is action tremor, which occurs during—and is triggered by—a voluntary action. Action tremor can be “postural” when tremor is triggered by the action of maintaining a position (zero speed) against gravity (postural tremor) or “kinetic” (with speed) when it is only triggered by movement (kinetic tremor) [2].

Essential tremor is a highly common action tremor. It is estimated to affect about 5% of the population over the age of 65 and 22% over the age of 95 [4,5]. Other action tremors may arise in all sorts of disorders involving cerebellar circuits [2,6], such as in multiple sclerosis where tremor is estimated to occur in 25% to 58% of cases [7,8,9]. Resting upper-limb tremor occurs in most patients with Parkinson’s disease, a condition estimated to affect 1% of individuals over the age of 60 [10,11]. Tremor that develops in a part of the body affected by dystonia may be called “dystonic tremor” [3].

Functional disturbances associated with tremor are profound: in a survey of more than 1500 patients with essential tremor, 89% reported difficulty with drinking, 85% with eating, and 75% with writing. In addition, 61% reported professional impact, with 13% having lost their job and 32% having changed their job [12]. Psychosocial hardship is also a serious issue: social phobia was found in 43% of participants and rates of antidepressant prescriptions were six times higher than in the general population [12]. In multiple sclerosis specifically, tremor is one of the most disabling symptoms, negatively affecting quality of life. Its severity correlates with disability and unemployment rate [7,13].

Given the above-mentioned frequency and functional impact of tremor, its treatment is of crucial importance. Avoidance of tremorogenic agents and a number of synaptic depressors has not been systematically evaluated to our knowledge. Promising physical techniques such as resistance or accuracy training programs have shown efficacy over very small series. This is an area of work that remains mostly unknown to the neurology community [14]. As for oral medications for essential tremor, there is a 30% non-response rate to drug treatment during the first year, and if treatment is pursued, a third of patients interrupt it during the second year due to lack of efficacy and/or adverse events (AEs) (notably sedation, falls with head trauma or fracture, cognitive impairment, bradycardia, syncope, severe anxiety, and suicidal ideation) [15,16]. It is worth noting that the two drugs that are still most commonly recommended (propranolol and primidone) were discovered in the 1970s and 1980s [15].

In Parkinson’s disease’s resting tremor, treatment is essentially based on dopaminergic agents, with the well-known fluctuations in efficacy and adverse events. Deep brain stimulation and functional lesions such as gamma-knife and focused ultrasounds may be highly effective but are also associated with risks. These procedures remain invasive and should be performed by well-trained teams [17].

In the past three decades, botulinum toxin injections have triggered growing interest as a treatment for tremor. Although patients might find this solution to be more satisfying than oral medications, it is still not widely offered in current practice. Indeed, 4% of patients with essential tremor benefit from botulinum toxin therapy in France, vs. 93% for systemic synaptic depressors such as propranolol, primidone, or benzodiazepines [12].

In this systematic review, we aimed to provide an update on the clinical trials that have focused on botulinum toxin injections to treat upper-limb tremor, and to discuss the perspectives to improve clinical practice.

## 2. Results

This systematic literature search resulted in 117 findings including 80 duplicates, yielding 37 publications. After screening, 21 manuscripts were excluded because they did not refer to tremor and/or did not target the upper limb or were not in the English or French language (Figure 1). After reviewing all 16 remaining manuscripts, 1 was excluded from the analyses because it was ultimately not considered relevant (a case series of two Parkinson’s disease and five essential tremor patients that was discussing the transition from a unilateral to a bilateral botulinum toxin injection pattern) [18]. In the end, 15 studies were collected for analysis.

### 2.1. Data from RCTs

Findings from RCTs are presented in Table 1 and Table 2. In July 2023, only eight RCTs were found in the literature regarding botulinum toxin injections to treat upper-limb tremor.

Four studies referred to essential tremor. Among them, two early trials used a fixed-dose, fixed-muscle approach (both omitting pronators and using only wrist flexor and extensor injection) with mitigated results, particularly in terms of functional performance. Indeed, the first study found improvement in tremor severity on a four-point patient-reported subjective scale and in accelerometry, but no improvement in functional rating scales. The second study found a dose-dependent improvement in a patient-reported subjective scale, but not in writing, working, social embarrassment, or anxiety [19,20]. The other two studies used case-customized methodology, potentially guided by kinematic analyses. They had more compelling results on tremor severity scores such as the Fahn Tolosa Marin Clinical Rating Scale for Tremor (FTM) or the National Institutes of Health Collaborative Genetic Criteria for tremor severity (NIHCGC), patient global impression of change (PGIC), or accelerometric tremor amplitude measurements up to 8 weeks after botulinum toxin injections [21,22]. Of note, a minority (n = 30) of the enrolled participants were pre-evaluated using kinematics to analyze agonist/antagonist muscle patterns before injection [22].

All studies reported on excessive weakness, even though one report claimed that this was no greater than in the placebo group (although it should be noted that in that study at least one patient was excluded from the analyses because of “excessive weakness”) [21]. One study found weakness to be more prevalent in the “high dose” group compared with the “low dose” group [20].

In multiple sclerosis, two RCTs evaluated injection effects according to a customized approach, targeting agonist and antagonist muscles depending on the tremor pattern: the Bain score was improved for up to 12 weeks, as were writing and Archimedes spirals, but quality-of-life scores were not [23,24]. Interestingly, in one of these studies, functional MRI activation within two previously identified areas was measured at baseline and 6 weeks: activation was reduced in the ipsilateral inferior parietal cortex in the BT group but not in the placebo group, with correlation with the reduction in tremor severity; activation was unchanged in the premotor/supplementary motor cortex [24]. Excessive weakness was again found in both multiple sclerosis studies [23,24].

Only one RCT evaluated the effects on Parkinson’s disease’s resting tremor, with an individualized injection pattern based on clinical judgement showing an improvement in all assessment criteria (UPDRS III items for resting tremor and for action tremor, NIHCGC, PGIC, and quality of life) for up to 8 weeks [25]. Weakness was detected using an ergometer in 37% of patients with Parkinson’s disease compared with 22% of control subjects [25]. Lastly, a single clinical study was carried out on “dystonic” tremor, using injections based on clinical judgement, with mixed results: improvement was found in the FTM total score but not in its functional sub-score, with no improvement in the writer’s cramp rating scale or in accelerometry performed on the index finger. Cases of troublesome weakness were found despite the low doses used [26].

Of the eight RCTs, three were cross-over studies with wash-out periods of 12 to 16 weeks that could be viewed as short, as BT—which has been particularly examined for abobotulinumtoxinA—may retain efficacy beyond 24 weeks [27,28]. A total of 351 subjects were included in these RCTs, including 133 in a study, of arguable real-life relevance, in which a fixed dose was injected only into the wrist flexors and extensors of patients with essential tremor [20]. Levels of Evidence ranged from 2.5 (a well-designed RCT, but with few patients and not multi-site) to 3 (questionable randomization) [29]. The CASP for RCTs was considered “moderate” for all studies, as recruitment was adequate, but the results could not be considered robust due to questionable methods and/or the limited number of subjects included [30]. 

### 2.2. Data from Open-Label Studies

Findings from the seven open-label studies selected are presented in Table 3 and Table 4. Among the seven articles, five investigated the effects of botulinum injections in essential tremor. One of these studies used a fixed-dose, fixed-muscle injection pattern in wrist flexors and extensors [31]. The other studies used a more personalized approach, with EMG or kinematic analyses [32,33,34,35]. Among those, three targeted muscles from wrist to shoulder (thus avoiding finger flexors and extensors) [33,34,35], and one deliberately avoided wrist extensors (in order to “avoid exaggerated weakness”) [32]. Interestingly, the published kinematic analyses used motion sensor devices including three goniometers and one torsiometer placed over the forearm, wrist, elbow, and shoulder joints during two postural tasks and two weight-bearing tasks; at the shoulder level, these kinematic analyses assessed shoulder flexion–extension and abduction–adduction, but it is not clear whether they also considered shoulder rotation [33,34,35]. All of these studies reported worthy improvements in essential tremor, on various severity scores (mainly FTM) and also on function (ADLS), accelerometry, and quality-of-life scores (QUEST). Improvement duration reached 96 weeks in one long-term study in which injections were repeated every 16 weeks [35]. All of these investigations also reported weakness, particularly affecting finger and wrist muscles.

The two remaining reports addressed mixed populations of patients with essential tremor, Parkinson’s disease, and “cerebellar disorders” [36,37]. One early study used an injection pattern that has become uncommon today, involving “booster” injections every 10–14 days for the first month, and reported subjective improvement in patients (PGIC) but no improvement in quantitative tests, with 60% of patients showing “excessive” weakness [36]. The other more recent report used a customized approach guided by kinematics, with improvement in all parameters (FTM, kinematics, and quality-of-life scores) but excluded a number of cases from the analyses—from 8 to 14%—because of “lack of improvement” or “bothersome weakness” [37].

Across these seven open-label studies, the cumulative number of patients included was 185. Ten were not counted because of redundancy in two studies [35,37]. Levels of Evidence ranged from 4 (well-designed cohort studies) to 6 (single descriptive or qualitative studies).

## 3. Discussion

This systematic review of the literature available since the emergence of botulinum toxins in the 1980s found few trials and even fewer RCTs published on the topic of botulinum toxin injections for treating upper-limb tremor. This review suggests that botulinum toxin is more effective when injections are individualized to the patient, as opposed to a fixed dose injected to fixed muscles. Such individualization may rely on the clinical judgement of an experienced clinician, on kinematic analysis, on surface electromyography, or on a combination of the above. Overall, subjective and objective improvements are often reported on tremor severity scores, functional scores, impressions of clinical improvement and quality-of-life scores, and also on quantitative kinematic measurements. However, transient and troublesome weakness remains a common issue after injection in these studies, and it is reasonable to assume that this is one of the reasons why injections for tremor are not more commonly used in current practice. We may also assume that another reason might be an impression of technical difficulty that such injections could give to some injectors.

The level of evidence is suboptimal as there are few published studies (eight RCTs, seven open-label studies), and a moderate cumulative number of subjects included (546 analyzed in this review) suffering from at least four different disorders. Methodological limitations included low sample sizes; lack of standardized methods for assessing efficacy and for measuring weakness, precluding rigorous comparisons between toxins, doses, and dilutions; lack of multicentric protocols (only in one study); and few intention-to-treat analyses, often with patients being excluded from the analyses because of injections deemed “ineffective” or because there was “exaggerated weakness”. In addition, only a few studies directly evaluated the overall patient’s point of view, as in the Patient Global Impression of Change (PGIC) [21,22,25,26]. This is unfortunate as self-detection of tremor in daily life activities may be more readily captured by such a tool than by other assessment methods.

### 3.1. Botulinum Toxin in Parkinson’s Resting Tremor

Resting tremor likely occurs as a result of central oscillatory activity, particularly in the basal ganglia under the influence of dopamine [6,38]. In this type of central dysfunctional pattern, it might be challenging to address tremor without weakening the muscle. Only one RCT has been conducted in patients with Parkinson’s disease [25]. Although findings were encouraging (Table 1), many clinicians remain hesitant to use botulinum toxin in Parkinson’s disease resting tremor due to uncertain efficacy and to the risk of induced weakness. It is important to remember that parkinsonian tremor often predominates distally in forearm muscles (wrist and fingers), and that, compared to proximal arm or shoulder muscles, induced weakness in these muscles may be readily perceived by the patient when injected with botulinum toxin. Yet, some specialists continue to perform botulinum toxin injections for Parkinson’s disease tremor, with interesting anecdotal results, and the field certainly deserves more high-quality investigations.

### 3.2. Botulinum Toxin in Action “Cerebellar” Tremors

When active posture or action is required, the cerebellum receives information about the anticipated movement from descending commands elaborated in the premotor cortices and transmitted along corticopontine pathways. In parallel, the cerebellum receives feedback from the ascending spinocerebellar tracts about the actual movements currently processed. These two pieces of information are compared within the cerebellar cortex and then corrected via the efferent cerebellar pathways [6,39,40]. When aiming at a target, this command correction is responsible for movement slowing or braking through contraction of the stretched antagonist muscle to avoid overshooting. Malfunction in these circuits can delay antagonist correction, which delays deceleration and results in hypermetria [39]. The cerebellum finally corrects the movement in the opposite direction to return to the target. Delays in antagonist corrections reiterate, and new reverse hypermetric movements are produced. As a result, there may be continuous overshooting and limb oscillation around the target, typical for ataxia or “cerebellar tremor” [39,40].

There is now growing evidence in the literature that “essential tremor” is likely a degenerative disorder predominantly affecting the cerebellum [41,42,43,44,45,46,47,48]. As reviewed above, feedback from the *muscle* plays an important part in fostering and maintaining this tremor, and a muscle-specific procedure such as botulinum toxin injection may reduce spindle afferent activity, regardless of contraction reduction [49] and may find its best indication there.

Botulinum toxin injected into muscle may restore reciprocal inhibition from the injected muscle to the non-injected muscle, through Renshaw blockade at the origin of the injected axon and through likely concurrent action on the extrafusal and intrafusal motor end-plates, the latter resulting in decreased spindle afferent input [49,50,51,52,53]. It is also likely that botulinum toxin exerts additional central actions, possibly relevant in tremors to damp oscillatory activities [54]. Since muscle afferent input influences central motor structures, such as the motor cortex, thalamus, and cerebellum, it is reasonable to assume that reduced input to these structures might lead to a reduction in oscillatory activity, and therefore, in tremor [31,49,55]. Recently, a study on tremor in multiple sclerosis demonstrated significant reduction in neural activation within the ipsilateral inferior parietal lobule in fMRI, after botulinum toxin injection [24]. Notably, the ipsilateral inferior parietal lobule plays a role in sensorimotor prediction and is a target of cerebellar efferents [56,57,58].

### 3.3. Botulinum Toxin in Dystonic Tremors

The pathophysiology of dystonic tremor remains unclear. Since the new syndromic understanding of the term dystonia promoted by Marsden and Fahn in the late 1970s [59,60], the scientific community has struggled to come up with a straightforward and universally accepted definition of dystonia [61,62]. In that context, it would appear that up to 50% of patients with dystonia develop alternating movements in the dystonic body area, a condition that has been termed dystonic tremor [63]. Given the established efficacy of BT injections in focal dystonia and in the treatment of other tremor types, BT for dystonic tremor has received surprisingly little attention from the scientific community, with only one published placebo-controlled trial to date (referred to in Table 1 and Table 2) [26]. The effectiveness of BT injections was particularly marked in that trial.

### 3.4. Going Further

The authors of this review believe that botulinum toxin offers strong promise for patients with tremor and should attract greater interest from clinicians and the scientific community.

#### 3.4.1. Comments and Suggestions for Best Practice

The following represents advice from the experience of the authors following this literature review:–Injecting finger muscles may cause troublesome weakness of grip and should be avoided whenever possible.–Movements of elbow pronation–supination seem to be those most often found in all activity tremors (essential tremors and other cerebellar tremors) and pronator muscles should thus not be omitted when planning for the injection.–Involvement of proximal muscles should not be underestimated, particularly in action tremors. Rotation movements of the shoulder are frequent in cerebellar tremors. Tests to assess this issue include bringing a glass from the table to the mouth, “finger-to-nose” movements, or the posture with elbows bent and fingertips close in the opposition. Unfortunately, there are no studies in the literature targeting these muscles, although anecdotal experience of the authors has shown promising results when targeting the pectoralis major, subscapularis, and/or teres major. In addition, patients with deep brain stimulation (DBS) often have a better outcome with respect to the distal components of tremor and may keep a persistent, troublesome proximal tremor, which could be a good indication for combining DBS with injections of botulinum toxin.–When deciding on the dose of botulinum toxin, caution should be used as weakness is dose-dependent [20].–To avoid diffusion to adjacent muscles, especially the muscles of the forearm and hands, it is better to not overdilute the toxin (e.g., one may use 100–200 U/mL for incobotulinumtoxin or onabotulinumtoxin; 300–500 U/mL for abobotulinumtoxin).–Doses: The total dose used per muscle and per injection session likely plays a role in the outcome but these differed among investigators. There are currently no consensus recommendations regarding the doses to be used in the treatment of the various types of tremor. From the overall literature in the field cited here and from our experience, we recommend using the following ranges of doses for a first injection, depending on the muscle size and trophic state: for abobotulinumtoxin, 60–100 U/forearm muscle, 140–200 U/arm or shoulder muscle, without exceeding 300–500 U/whole of a single upper limb; for incobotulinumtoxin or onabotulinumtoxin, 30–50 U/forearm muscle, 70–100 U/arm or shoulder muscle, without exceeding 150–200 U/whole of a single upper limb.–Injection techniques: As each has its advantages and limitations, a combination of the following BT injection techniques may be an ideal approach:
–Electromyography-guided injection makes it possible to hear the bursts of muscle contraction when the tremor occurs [64,65];–Electrical stimulation is precious as this is the sole technique ensuring that the *functional* effect of the stimulated muscle indeed corresponds to the tremor movements; however, it can be difficult to distinguish between contractions due to stimulation and those due to the tremor [64,65];–Ultrasound helps in ensuring that the needle is in the targeted muscle and sometimes makes it possible to see the contraction during the tremor, but it does not make it possible to know if the BT is injected to a muscle area actually causing the tremor movements, which could mitigate its effectiveness [66].


Using anatomical landmarks is the least effective strategy, as it has been shown to be overall less effective than the above-mentioned techniques [64]. Finally and importantly, the success of each of these techniques is highly dependent on the experience and skill of the clinician using it; therefore, all the published comparisons have been biased by the respective skills of the individual investigators in mastering each of the techniques [64,65,66].

#### 3.4.2. An “Asymmetric” Injection Pattern?

Currently, both in clinical practice and in the literature, botulinum toxin injections often target agonist *and* antagonist muscles. When injections are guided by kinematic analysis or surface electromyography, injections tend to be more “asymmetric” (targeting the flexor/pronator more than the extensor/supinator compartment), although studies to date have not established this strategy [67,68].

The authors of this review inject botulinum toxin asymmetrically, targeting agonist pronators or flexors *only*, in the forearm, as extensors or supinators might be particularly disabling when they become weaker [67,68]. Following the above-mentioned concept of the muscle acting in a neuro-muscular loop both as the origin of sensory afferents towards antagonist motoneurons and as the motor effector, it does not seem necessary to interrupt the loop from both sites (agonist and antagonist). Obviously, this concept of “asymmetric” injections needs to be demonstrated in well-designed clinical trials.

#### 3.4.3. Neurorehabilitation Techniques as an Adjunct to Botulinum Toxin Injections

Neurorehabilitation techniques are currently not widely used for tremor. Yet, when patients with essential tremor are asked the question: “If you were to design a comprehensive approach/ideal clinical center for the treatment of tremor, what problems other than tremor would you focus on?”, the second most common answer is “physiotherapy or occupational therapy in order to help with self or personal care or personal hygiene” (29%) [16]. Considering the “potential importance of specialists in the treatment of patients with essential tremor”, 62% of respondents considered the physiatrist or physical therapist to be “very important” or “essential” [16].

We here suggest that adding a neurorehabilitation protocol to toxin injections may prove useful, particularly for patients with essential or other cerebellar types of tremor. Two rehabilitation techniques have shown promise in that population: accuracy training and motor-strengthening techniques. Accuracy training is effective for cerebellar disorders, including essential tremor [69,70,71]. Simple accuracy exercises performed regularly, even daily, as part of a self-rehabilitation program could significantly improve movement control if more widely prescribed [72]. Motor-strengthening programs have also produced encouraging results in essential tremor and certainly deserve to be investigated in larger controlled trials [14,73,74,75,76]. Evidence for the rehabilitation of dystonic tremor is still lacking.

Taking all these aspects into account, much still needs to be done to improve knowledge about BT injections for tremor. There is a need for some form of uniform upper-limb tremor botulinum injection protocol usable for high-quality RCTs, with the objective to improve muscle targeting and the selection of dose and dilution to maximize efficacy and avoid exaggerated weakness, while ensuring an individualized approach for each patient. Improving these aspects could involve the use of kinematics. By analyzing angular movements applied to each joint in different tasks, these tools may allow objective identification of the muscles to be targeted, as well as tremor monitoring after BT injection [33,34,35]. This could be supported by automated algorithms, albeit under investigator control. While visual assessment remains key and the experience of the examiner plays a major role, they may be enhanced by kinematics for diagnostic accuracy and objectivity.

## 4. Methods

The research methodology followed the Preferred Reporting Items for Systematic Reviews and Meta-Analyses (PRISMA) guidelines [77].

### 4.1. Research Methodology and Inclusion Criteria

A search was carried out on PubMed until July 2023 with the following terms: “botulinum toxin” AND “tremor” AND/OR “essential tremor”, “Parkinson”, “multiple sclerosis”, “Wilson”, “dystonic”, “rubral”, “writing”, “hand”, “upper limb”, “proximal”, “distal”, “cerebellar”, and “action”. Article type was filtered to retain all “clinical trials” and “randomized clinical trials” (RCTs). Duplicates were removed by the primary investigator (DM) using the “sorting” function in Excel software (2016, v.16.0).

### 4.2. Screening Methodology and Exclusion Criteria

Two investigators (DM and CD) independently screened all titles and abstracts of the remaining studies. They excluded irrelevant articles, i.e., those not pertaining to botulinum toxin injections in upper limbs, those not intended to treat tremor of any type, and those that were not in the English or French language. Careful reading of all the selected studies was then carried out jointly by the same two investigators. Articles could still be excluded if the content was considered irrelevant.

### 4.3. Data Collection

The following data were collected and entered into an Excel database: first author, journal, year of publication, type of study (open-label or RCT), disease studied, number of groups, type of analyses (cross-over or longitudinal, intention-to-treat or per protocol), number of patients (total and per group), presence of placebo, botulinum toxin used, method of muscle selection, dose used, all reported outcomes including the clinical scales used, adverse effects (weakness and others), and identified bias.

### 4.4. Quality Assessment Method

The methodological quality of each study was assessed using the Levels of Evidence and the Critical Appraisal Skills Programme (CASP) checklist for RCTs [29,30]. Concerning CASP, quality assessments were categorized as “low” (no appropriate recruitment), “moderate” (appropriate recruitment but low or intermediate results), or “high” (appropriate recruitment and strong results). Any disagreements regarding screening or quality assessment could be settled by discussion with a third investigator (ER).

## Figures and Tables

**Figure 1 toxins-16-00392-f001:**
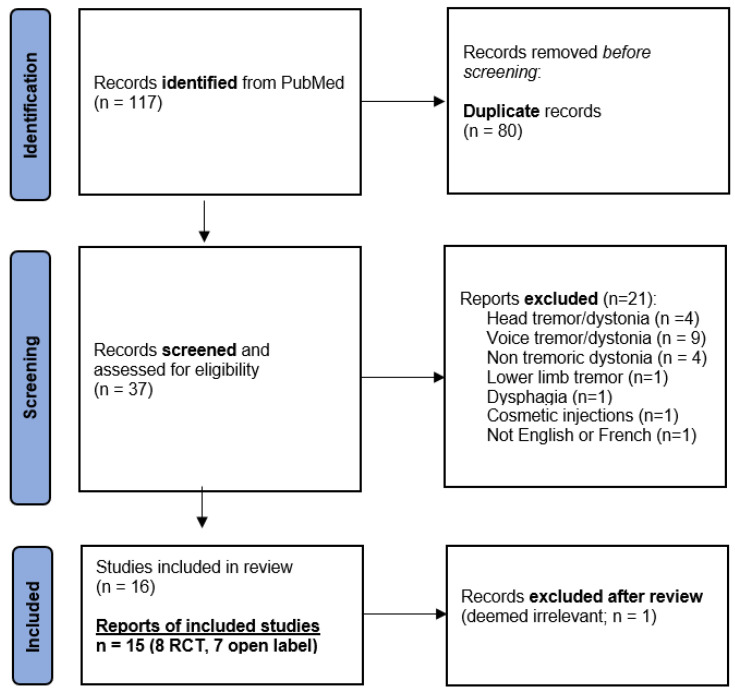
Flowchart of study selection.

**Table 1 toxins-16-00392-t001:** Randomized controlled trials—part 1.

** *Essential Tremor (n = 4)* **							
** *Ref.* **	** *n* **	** *BoNT* **	** *Muscle Selection* **	** *Dosage & Dilution* **	** *Results* **	** *Weakness* **	** *Other AE* **
Jankovic J. *Mov. Disord.* *1996* [19]	**25** *-13 BT* *-12 placebo*	Ona-A	**Fixed:** Wrist F/E	**Fixed:** 50 U +/− repeated with 100 U (if no efficacy in week 4) *Dilution: 100 U/mL*	**Improvements compared to placebo on:** -Severity tremor scale (91.7% in BT, 36.4% in placebo; *p* = 0.009 at W4) -Accelerometry (median peak amplitude −10/−20% from W4 to W12) **No improvements on:** -Functional rating scales and EMG responses	**Yes** -Finger weakness -All TB patients (50% mild and 50% moderate)	No
Brin M.F. *Neurology* *2001* [20]	**133** *-43 low-dose* *-45 high dose* *-45 placebo*	Ona-A	**Fixed:** Wrist F/E	**Fixed:** “Low-dose” 50 U “High-dose” 100 U *Dilution:* *50 U/mL (low dose)* *100 U/mL (high dose)*	**Improvements compared to placebo on:** -Severity tremor scale at W6-12-16 for postural tremor and at W6 for kinetic tremor (high-dose and low-dose groups) -On spiral and straight-line drawings at W6 (high-dose and low-dose) -On pouring water at W16 (high-dose) -On functional rating scales: feeding, dressing and drinking (high-dose and low-dose) and on hygiene and fine movements (high-dose) **No improvements on:** -Handwriting, speaking, working, embarrassment or anxiety	**Yes** -Wrist weakness (dynamometer and clinical) -30% of “low-dose” patients, 70% in “high-dose”	High-dose group: -3 paresthesias
Mittal S.O. *Parkinsonism* *Relat. Disord.* *2018* [21]	**33** *-Cross-over*	Inco-A	**Customized:** Fingers, wrist or elbow including P/S (clinical judgement)	**Customized:** Mean total dose 100 U *Dilution: 100 U/mL*	**Improvements compared to placebo on:** -FTM tremor severity score at W4 and W8 -NIHCGC tremor severity scale at W4 and W8 -Patient’s global impression of change	**No** -Not significant compared to placebo (BT group n = 6, placebo group n = 4, *p* = 0.7)	No
Jog M. *Toxins* *2020* [22]	**30** *-19 BT* *-11 placebo* *(2:1 randomization)*	Inco-A	**Customized:** Kinematics with algorithm +/− modified by investigator	**Customized:** Total dose 30–200 U *Dilution: not specified*	**Improvements compared to placebo:** -FTM motor performances score at W4 and W8 -Patient’s global impression of change at W4 -Accelerometric hand-tremor amplitude at W4 and W8	**Yes** -”Slight and transient” on maximal grip strength -2 patients on “finger extension”	BT group: -1 dry mouth -1 dysphonia
** *Multiple Sclerosis (n = 2)* **							
Van Der Walt *Neurology* *2012* [23]	**23** *-Cross-over* *-33 upper limbs*	Ona-A	**Customized:** Agonist and antagonist muscles relating to tremor pattern (F/E of wrist and elbow and P/S)	**Customized:** Max total dose 100 U Mean total 83 U, range 35–100 *Dilution: 50 U/mL*	**Improvements compared to placebo on:** -Bain score for tremor severity (kinetic and postural), writing and Archimedes spiral at W6-12 **No improvements on:** -Quality of life.	**Yes** -Mild to moderate -42.2% of patients with BT, 6.1% with placebo (*p* < 0.001)	No
Boonstra F. *Mult. Scler.* *Relat. Disord.* *2020* [24]	**43**	Ona-A	**Customized** (clinical judgment)	**Customized:** Max total dose 150 U *Dilution: 50 U/mL*	**Improvements compared to placebo on:** -Bain score for handwriting at W6 and W12 and for tremor severity at W12 -Reduction in activation within the ipsilateral inferior parietal cortex (measured only at W6) which correlated with the reduction of tremor **No improvements on:** -Reduction in activation in the premotor/supplementary motor cortex	**Yes** -Significant decrease in strength (MRC) compared to placebo at W6 (but not at W12)	No
** *Parkinson’s Disease (n = 1)* **							
Mittal S.O. *Mayo. Clin. Proc.* *2017* [25]	**34** *-Cross-over*	Inco-A	**Customized:** Fingers, wrist or elbow including P/S (clinical judgement)	**Customized:** Mean total dose 100 U *Dilution: 100 U/mL*	**Improvements compared to placebo on:** -UPDRS item 20 (rest tremor) at W4 and W8 -UPDRS item 21 (postural/action tremor) at W8 -NIHCGC tremor severity scale at W4 and W8 -Patient’s global impression of change at W4 and W8 -Quality of life (PDQL) at W4 and W8	**No** -Not significant compared to placebo (BT group 37%, placebo group 22%) (assessed by ergometer)	No
** *Dystonic Tremor (n = 1)* **							
Rajan R. *JAMA Neurol.* *2021* [26]	**30** *-15 BT* *-15 placebo*	Ona-A	**Customized** (clinical judgment)	**Customized:** Mean total dose 63 ± 29 U *Dilution: 50 U/mL*	**Improvements compared to placebo on:** -FTM total score at W6 and W12 -FTM for tremor severity and motor performances subscores at W6 -Patient’s global impression of change **No improvements on:** -FTM for functional disability -Writer’s cramp rating scale -Accelerometric measures (only done on index finger)	**Yes** -1 severe (wrist drop) -40% complaint in BT group and 29% in placebo group -No difference in grip strength (dynanometer)	No

**Table 2 toxins-16-00392-t002:** Randomized controlled trials—part 2.

** *Essential Tremor (n = 4)* **						
** *Ref.* **	** *Selection Criteria* **	** *Guidance Technique* **	** *Injected Muscles* **	** *Risk of Bias* **	** *Evidence* **	** *CASP* **
Jankovic J. *Mov. Disord.* *1996* [19]	-“Typical ET” -≥21 yo -No other cause of tremor -Not previously treated with BT	Anatomical landmarks	Flexor carpi radialis Flexor carpi ulnaris Extensor carpi radialis Extensor carpi ulnaris	BT group: -Greater male representation	2.5	Moderate
Brin M.F. *Neurology* *2001* [20]	-ET as defined by Tremor Investigation Group criteria -Hands bilateral postural tremor -Severity ≥2 on a four-point scale -Not previously treated with BT -No current use of antitremor medications	EMG	Flexor carpi radialis Flexor carpi ulnaris Extensor carpi radialis Extensor carpi ulnaris	-	2.5	Moderate
Mittal S.O. *Parkinsonism* *Relat. Disord.* *2018* [21]	-“Clinical diagnosis of ET with moderate to severe tremor” -≥18 yo -Unchanged medications and/or non-pharmacological treatments	EMG	Not precisely specified (from hand to elbow)	-Short washing-out period (cross-over at W16) -Withdrawn = 5 (1 excessive weakness)	2.5	Moderate
Jog M. *Toxins* *2020* [22]	-ET ≥6 months -Stability ≥4 weeks -Diagnosis of “definite ET” (Tremor Investigation Group criteria) -Moderate-to-marked upper-limb postural and/or kinetic tremor at wrist level (FTM Part C) -Unchanged anti-tremor medication	Ultrasound, EMG and/or electrical stimulation (determined by the investigator)	Mandatory in ≥2 wrist muscles: Flexor carpi radialis Flexor carpi ulnaris Extensor carpi radialis Extensor carpi ulnaris Pronator quadratus Pronator teres Supinator Optional in elbow and/or shoulder	Withdrawn = 3 -1 in BT (lost) -2 in placebo	2.5	Moderate
** *Multiple Sclerosis (n = 2)* **						
Van Der Walt *Neurology* *2012* [23]	-Relapsing remitting and secondary progressive MS with disabling arm tremor -No BT within last 6 months -Antitremor medications ceased 1 week before study	EMG +/− electrical stimulation (if needed)	All muscles (from shoulder to fingers) were injected depending on the case (see Table 1 in the cited article).	-Short washing-out period (cross-over at W12) -Withdrawn = 2 upper limbs (MS relapses)	3	Moderate
Boonstra F. *Mult. Scler.* *Relat. Disord.* *2020* [24]	-Relapsing-remitting and secondary-progressive MS with unilateral upper limb tremor -No BT within last 6 months -Antitremor medications ceased 1 week before study -Without upper limb weakness	With “EMG and stimulation device”	Not specified	-Withdrawn = 3 (unknown)	2.5	Moderate
** *Parkinson’s Disease (n = 1)* **						
Mittal S.O. *Mayo. Clin. Proc.* *2017* [25]	-Clinical diagnosis of PD with moderate to severe tremor refractory to standard medical treatments -≥18 yo -No BT within last 4 months -Unchanged medications and non-pharmacologic treatments during study	EMG	All muscles (from elbow to fingers) were injected depending on the case (see Table 1 in the cited article).	-Short washing-out period (cross-over at W12) -Withdrawn = 4	2.5	Moderate
** *Dystonic Tremor (n = 1)* **						
Rajan R. *JAMA Neurol.* *2021* [26]	-Isolated focal, segmental, multifocal, or generalized dystonia and brachial dystonia with dystonic tremor as defined by the MDS Consensus criteria for tremor (Axis I) -≥18 yo -No BoNT within 4 months -Stable dose of medications for tremor within last month	EMG	All muscles (from elbow to fingers + “deltoid”) were injected depending on the case (see Table 1 in the cited article).	-Withdrawn= 1 in BT group, 4 in placebo but intention-to-treat analyzis	2.5	Moderate

**Table 3 toxins-16-00392-t003:** Open-label studies—part 1.

** *Essential Tremor (n = 5)* **							
** *Ref.* **	** *n* **	** *BoNT* **	** *Muscle Selection* **	** *Dosage & Dilution* **	** *Results* **	** *Weakness* **	** *Other AE* **
Modugno N. *Muscle Nerve* *1998* [31]	**20** *-10 ET* *-10 control*	Ona-A	**Fixed:** Wrist F/E	**Fixed:** Total 80–100 U *Dilution: 50 U/mL*	**Improvements on:** -Modified FTM for functional disability at W4 -Reciprocal inhibition (reduced second presynaptic phase in ET, which is normalized after BT injection)	**Yes** -Mild wrist weakness -60% of ET patients	No
Pacchetti C. *Neurol. Sci.* *2000* [32]	**20**	Abo-A	**Customized:** -EMG assessments and clinical judgment -Wrist extensors not injected to avoid weakness (unless they were the only ones involved)	**Customized:** Mean total 95 ± 40 U Mean by muscle 68 ± 26 U *Dilution: 200 U/mL*	**Improvements on:** -Severity tremor scale at W4 and W12 -ADLS at W4 and W12 -Mean tremor amplitude on accelerometry at W4 and W12 -Mean amplitude of contraction on EMG during the two most disabled tasks at W4 and W12 -All improvements were no more found at W20	**Yes** -Finger extension weakness -15% of patients (only ones injected in extensor carpal radialis)	No
Samotus O. *PLoS ONE* *2016* [33]	**24**	Inco-A	**Customized:** -Kinematics and clinical judgment -From wrist to shoulder (but not shoulder rotation)	**Customized:** -Injections at W0, W16 and W32 -Adapted depending on improvement since last injection *Dilution: 200 U/mL*	**Improvements starting W6 and until W38 on:** -FTM for tremor severity, writing and functional ability -Quality of life (QUEST) -Objective kinematic assessments	**Yes** -Mild and not interfering with arm function -40% of patients -Mean difference in grip strength between W0 and W6 = 6.51 ± 1.54 kg -1 withdrawn because of “unwanted weakness”	No
Samotus O. *Toxins* *2019* [34]	**31**	Inco-A	**Customized:** -Only kinematics with automated algorithm -From wrist to shoulder (but not shoulder rotation)	**Customized:** -3 bilateral injections during 30 weeks -On initial injection: Mean total 136 ± 54 U on predominant side Mean total 125 ± 62 U on non-dominant *Dilution: not specified*	**Improvements in both sides on:** -FTM for tremor severity and functional ability -Quality of life (QUEST) -Objective kinematic assessments	**Yes** -17.6% finger weakness -1 patient had mild elbow weakness -No patient had shoulder weakness	No
Samotus O. *Can. J. Neurol. Sci.* *2018* [35]	**10**	Inco-A	**Customized:** -Kinematics and clinical judgment -From wrist to shoulder (but not shoulder rotation)	**Customized:** -Injections every 16 weeks and follow during 96 weeks -On initial injection: Mean total 185 ± 37 U Mean by muscle 10 ± 2 U -Adapted depending on improvement since last injection *Dilution: 200 U/mL*	**Improvements on:** -FTM for tremor severity and functional ability -Quality of life (QUEST) -UPDRS item 21 (postural/action tremor) -Objective kinematic assessments	**Yes** -Moderate -70% of patients at W54 -Mean grip strength at W0 27 ± 12 versus 20 ± 16 at W6	No
** *Mixed population (n = 2)* **							
Pullman S.L. *Arch. Neurol.* *1996* [36]	**38** *-15 PD* *-17 ET* *-5 Cerebellar* *(among 187 injected for all “limb disorders”)*	Ona-A	**Customized:** Methods not detailed	**Customized:** -Clinical judgment +/− “booster injections” every 10-14 days during first month *Dilution: 25, 50 or 100 U/mL*	**Improvements on:** -Patient’s global impression of change (13.3% PD; 17.6% ET; not expressed for cerebellar tremors) **No improvements on:** -Quantitative measures	**Yes** -Finger and wrist weakness -All TB patients -59.5% of “excessive” weakness	No
Samotus O. *PLoS ONE* *2017* [37]	**52** *-28 PD* *-24 ET*	Inco-A	**Customized**: -Kinematics an clinical judgment -From wrist to shoulder (but not shoulder rotation)	**Customized:** -Injections every 16 weeks and follow during 96 weeks -On initial injection: Mean total 147 ± 69 U in PD; Mean by muscle 8 ± 2 U in PD; Mean total 169 ± 63 U in ET Mean by muscle 9 ± 2 U in ET *Dilution: 200 U/mL*	**Improvements on:** -FTM for tremor severity and functional ability in PD and ET -Quality of life only in ET -Objective kinematic assessments in PD and ET	**Yes** -“Mild and not interfering with arm function” and only before W22 -But withdrawn = 14% in PD and 8% in ET because of “bothersome weakness”	No

**Table 4 toxins-16-00392-t004:** Open-label studies—part 2.

** *Essential Tremor (n = 5)* **					
** *Ref.* **	** *Selection Criteria* **	** *Guidance Technique* **	** *Injected Muscles* **	** *Risk of Bias* **	** *Evidence* **
Modugno N. *Muscle Nerve* *1998* [31]	-Diagnosis of ET based “on clinical criteria” -No other neurological abnormalities -No drugs known to cause tremor -Antitremor drugs stopped 5 days before study	Anatomical landmarks or EMG	Flexor carpi radialis Flexor carpi ulnaris Extensor carpi radialis Extensor carpi ulnaris	-	4
Pacchetti C. *Neurol. Sci.* *2000* [32]	-Diagnosis of ET based “on clinical criteria” -No other neurological abnormalities -No recent tremorogenic drugs or other secondary causes of tremor	“Without EMG guidance” (presumably anatomical landmarks)	Flexor carpi radialis Extensor carpi radialis Biceps Triceps	-	4
Samotus O. *PLoS ONE* *2016* [33]	-Diagnosed with ET with upper limb tremor -18−80 yo -Not previously treated with BT -No medication change within last 6 months -No tremorogenic drug	EMG	Flexor carpi radialis Flexor carpi ulnaris Extensor carpi radialis Extensor carpi ulnaris Biceps Triceps Pectoralis major Teres major Supraspinatus	-Single investigator -Withdrawn = 3 1 unrelated event 1 failed attendance 1 “unwanted weakness”	4
Samotus O. *Toxins* *2019* [34]	-ET as defined by Tremor Investigation Group criteria -Tremor in both upper limbs -No medication change within last 3 months -Not previously treated with BT -No tremorogenic drug	EMG	Not specified	-Withdrawn = 8 4 for insufficient perceived benefit and/or weakness	4
Samotus O. *Can. J. Neurol. Sci.* *2018* [35]	-Diagnosed with ET with upper limb tremor -18−80 yo -Not previously treated with BT -Naive of any oral antitremor medication -No tremorogenic drug	EMG	All muscles (from shoulder to fingers) depending on the case (see Table 1 in the cited article)	-Part of 2 already published articles [33,37]: only ET patients naive of medication and toxin were taken into account -Single investigator -Withdrawn = 2 1 renal failure 1 not clear	4
** *Mixed population (n = 2)* **					
Pullman S.L. *Arch. Neurol.* *1996* [36]	-Prospective survey of “patients with limb disorders”	EMG	All muscles (from shoulder to fingers) depending on the case (see Table 3 in the cited article)	-Too many to be listed	6
Samotus O. *PLoS ONE* *2017* [37]	-Upper limb tremor with diagnosis of ET (Tremor Investigation Group criteria) or idiopathic PD (UK Brain Bank Criteria) -Hoehn&Yahr stages 1–3) -18–80 yo -Not previously treated with BT -No medication change within last 6 months -No tremorogenic drug	Presumably EMG (unclear)	Not specified	-Single investigator -Many whitdrawns = -in PD: 14% bothersome weakness, 11% lack of improvement -in ET: 8% bothersome weakness, 8% lack of improvement	4

## Data Availability

All available data are provided in the figure and tables published in this manuscript.

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
