# Peer review of "Use of Botulinum Toxin in Upper-Limb Tremor: Systematic Review and Perspectives"

_toxins, 2024, doi:10.3390/toxins16090392_

Round 1
Reviewer 1 Report
Comments and Suggestions for Authors
Overall well written paper of clinical significance.
No major methodological concerns.
I would appreciate some comments of targeting techniques such as use of ultrasound and electrical stimulation, which are very relevant when targeting forearm muscles, frequently involved in UE tremor.
Comments on the Quality of English LanguageMinor language editing required, overall well written in terms of grammar, syntax and style.
Author Response
Overall well written paper of clinical significance.
No major methodological concerns.
I would appreciate some comments of targeting techniques such as use of ultrasound and electrical stimulation, which are very relevant when targeting forearm muscles, frequently involved in UE tremor.
Thank you for the comments. Discussing injection techniques is indeed essential, even though no controlled study has compared them in tremor to our knowledge and results of comparisons are all biased by the individual skills of the investigators in mastering this or that technique. We have now added the following text in the Advice based on personal experience paragraph (lines 345-361):
- Injection techniques: As each has its advantages and limitations, a combination of the following BT injection techniques may be an ideal approach:
--electromyography-guided injection makes it possible to hear the bursts of muscle con-traction when the tremor occurs;
--electrical stimulation is precious as this is the sole technique ensuring that the functional effect of the stimulated muscle indeed corresponds to the tremor movements; however, it can be difficult to distinguish between contractions due to stimulation and those due to the tremor;
--ultrasound helps ensuring that the needle is in the targeted muscle and sometimes makes it possible to see the contraction during the tremor, but it does not make it possible to know if the BT is injected to a muscle area actually causing the tremor movements, which could mitigate its effectiveness.
Using anatomical landmarks is the least effective strategy, as it has been shown to be overall less effective than the above-mentioned techniques [65]. Finally and importantly, the success of each of these techniques is highly dependent on the experience and skill of the clinician using it, therefore all the published comparisons have been biased by the respective skills of the individual investigators in mastering each of the techniques [65-67].
Reviewer 2 Report
Comments and Suggestions for Authors
This is a review of the treatment of tremor and we felt it was of great academic importance. Indeed, as noted in the paper. In a survey of over 1500 patients with essential tremor, 89% had difficulty drinking, 85% had difficulty eating and 75% had difficulty writing. In a survey of 1500 patients with essential tremor, 89% reported difficulty drinking, 85% eating and 75% writing. However, although the main focus is on botulinum therapy for tremor, there are no specific examples to aid treatment, such as how many units were administered to what specific muscles. It would be better if some kind of presentation in this respect was necessary and would make the literature more meaningful as it would give an idea of the treatment methods.
Author Response
This is a review of the treatment of tremor and we felt it was of great academic importance. Indeed, as noted in the paper. In a survey of over 1500 patients with essential tremor, 89% had difficulty drinking, 85% had difficulty eating and 75% had difficulty writing. In a survey of 1500 patients with essential tremor, 89% reported difficulty drinking, 85% eating and 75% writing. However, although the main focus is on botulinum therapy for tremor, there are no specific examples to aid treatment, such as how many units were administered to what specific muscles. It would be better if some kind of presentation in this respect was necessary and would make the literature more meaningful as it would give an idea of the treatment methods.
ANSWERS:
Thank you for the comments. Unfortunately, there are currently no consensus recommendations regarding the doses to be used in the treatment of the various types of tremor. Total doses used per muscle and per injection session vary amongst investigators. An effort has been made to specify the exact doses used in the studies cited in this review (see Tables 1 & 3) and we have added the following text (lines 336-344):
Total doses used per muscle and per injection session likely play a role in the outcome but they differ among investigators. There are currently no consensus recommendations regarding the doses to be used in the treatment of the various types of tremor. From the overall literature in the field cited here and from our experience, we recommend using the following ranges of doses for a first injection, depending on the muscle size and trophic state: for abobotulinumtoxin 60-100 U/forearm muscle, 140-200 U/arm or shoulder muscle, without exceeding 300-400 U/whole of a single upper limb; for incobotulinumtoxin or onabotulinumtoxin 30-50 U/forearm muscle, 70-100U/arm or shoulder muscle, without exceeding 150-200 U/whole of a single upper limb.
Reviewer 3 Report
Comments and Suggestions for Authors
The topic of the paper is relevant. Treatment of upper limb tremor is far from being standardized and the use of Botulinum Toxin (BoNT) is not always taken into account. Furthermore, BoNT treatment for tremor, as highlighted in the paper, could be less considered due to an impression of technical difficulty. The review aims to provide the status regarding trials of BoNT treatment for upper limb tremor.
The quality of the review is overall satisfactory but needs major revisions.
The “Introduction” paragraph lacks an explanation of what dystonic tremor is since one of the analyses subsequently performed is carried out also in this specific type of tremor.
The search strategy and the inclusion criteria are appropriate, and the criteria used for the selection of the reports appear to be valid. The screening process was done by two investigators and reviewed by a third investigator.
Selected studies have been grouped in randomized clinical trials (RCT) and open-label (OL) and these are assessed thoroughly one by one.
Which were the criteria used for the selection of the population of tremor patients in the methods of the different RCT studies need to be analyzed further.
I suggest clarifying the aspects where the authors of the first study cited found an improvement (line 116).
Fig 1 about search strategies: I will specify at the end that 15 selected papers are 8 RCT and 7 OS
Line 130: a dot is missing.
Tables: I would suggest adding some columns because some important points have to be highlighted. Do you have analyzed in the studies the dilution of the injected toxins? Is there any correlation between dilution and side effects or outcomes? did you find in the studies more weaknesses in using any particular toxin or dilution? which was in the 8 RCT the exact time of evaluation of the effect? Did the authors use any guide during injections? did you find any common criteria in the selection of the muscles and the injection technique? How many patients were pre-evaluated (polYgraphy, sensors, kinematics..) to analyze agonist/ antagonist muscle patterns before injections? Which are the medium/maximum dosages used in the studies for the different toxins? How many patients received only one injection and how many had a follow-up with repeated injections and for how long? Did Herz's measurement of tremor was performed in some studies?
I suggest adding a new table of mostly injected muscles.
All these results could definitively add value to the discussion.
Line 147 please be clear about which paper reports (add REF) that toxin is effective up to 24 weeks, this is not typical clinical practice in BoNTA clinic.
When discussing the results of the systematic review I would stress the lack of a standardized and objective method of weakness assessment and that the studies selected used different assessing scales for primary outcomes.
How many studies consider the patient's point of view (pGIC)? is this relevant in the auto-detection of tremor in daily life activities? Probably this point has to be also stressed in the discussion.
I consider too long the description of the pathogenesis of different tremor syndromes, even if interesting, in particularly for the cerebellar tremor. The mechanism of tremor is important considering the possible effect of BoNTA but has to be synthesized; the mechanism of dystonic tremor is also to be taken into account underlying why it could be more beneficial of BoNT treatment in comparison with other types of tremor (ie cerebellar tremor due to lesions or neurodegeneration).
The “Going further” paragraph lists and summarizes advice from the authors, so I would underline that these are opinions, possibly expert opinions, but they are not properly “lessons learned from this review”(line 255). The results of this review process needed to be underlined to be useful for the scientific community.
Till now there is a lack of evidence of the efficacy of the rehabilitation approach in tremors. I will suggest reformulating and shortening this paragraph and clarifying the differences in the possible retraining techniques in different typologies of tremor. Anyway, this is not the main topic of this review.
Regarding motor strengthening programs (line 300) it should be noted that not all tremor types would benefit from this kind of rehabilitation, ie dystonic tremor.
Citations are not completely adequate, papers on spasticity are cited several times with no particular congruity; furthermore, there is lacking evidence in the literature and in clinical daily life that tremor has to be treated with similar strategies.
Cit 68-69-70-71 is not fully related to this subject.
Lastly, I would strongly highlight for the future the need for an upper limb tremor uniformed botulinum injection protocol, to start RCTs without too strong methodology issues.
no main problems
Author Response
The topic of the paper is relevant. Treatment of upper limb tremor is far from being standardized and the use of Botulinum Toxin (BoNT) is not always taken into account. Furthermore, BoNT treatment for tremor, as highlighted in the paper, could be less considered due to an impression of technical difficulty. The review aims to provide the status regarding trials of BoNT treatment for upper limb tremor.
The quality of the review is overall satisfactory but needs major revisions.
The search strategy and the inclusion criteria are appropriate, and the criteria used for the selection of the reports appear to be valid. The screening process was done by two investigators and reviewed by a third investigator. Selected studies have been grouped in randomized clinical trials (RCT) and open-label (OL) and these are assessed thoroughly one by one.
Comments on the Quality of English Language: no main problems
ANSWERS:
We thank the Reviewer for careful consideration of our article. The Reviewer raises interesting points. We have tried to answer your questions as fully as possible.
COMMENT 1: The “Introduction” paragraph lacks an explanation of what dystonic tremor is since one of the analyses subsequently performed is carried out also in this specific type of tremor.
We have now added a statement in that respect in Introduction (lines 37-40):
Tremor that develops in a part of the body affected by dystonia may be called 'dystonic tremor', a type of action tremor sometimes described as kinetic (such as in task-specific tremor), postural or isometric (during muscle contraction against a rigid stationary object) [3].
COMMENT 2: Which were the criteria used for the selection of the population of tremor patients in the methods of the different RCT studies need to be analyzed further.
The response to this question has been pooled with that to Comment 6 (see below).
COMMENT 3: I suggest clarifying the aspects where the authors of the first study cited found an improvement (line 116).
Clarification now added lines 140-146:
Among them, two early trials used a fixed-dose fixed-muscle approach (both omitting pronators and using only wrist flexor and extensor injection) with mitigated results, particularly in terms of functional performance. Indeed, the first study found improvement in tremor severity on a 4-point patient-reported subjective scale and in accelerometry, but no improvement in functional rating scales. The second study found a dose-dependent improvement in a patient-reported subjective scale, but not in writing, working, social embarrassment or anxiety [22,23].
COMMENT 4: Fig 1 about search strategies: I will specify at the end that 15 selected papers are 8 RCT and 7 OS
The modification has been made (Fig 1).
COMMENT 5: Line 130: a dot is missing.
Corrected.
COMMENT 6: Tables: I would suggest adding some columns because some important points have to be highlighted.
-Do you have analyzed in the studies the dilution of the injected toxins?
This point has now been added in Tables 1 & 3, column “Dosage & Dilution”.
-Is there any correlation between dilution and side effects or outcomes?
This could not be reliably assessed as there were no comparable pairs of trials using similar doses with different dilutions.
-did you find in the studies more weaknesses in using any particular toxin or dilution?
Again, this is difficult to ascertain as each study used a single toxin (specified in the ‘BoNT’ column of Tables 1 & 3), with no head-to-head comparisons between toxins. In addition, assessments of outcomes and adverse effects were not standardized across studies.
We have added a comment in that respect at the end of the paper, lines 395-400:
Taking all these aspects into account, much still needs to be done to improve knowledge about BT injections for tremor. There is a need for some form of upper limb tremor uniform botulinum injection protocol usable for high-quality RCTs, with the objective to improve muscle targeting and the selection of dose and dilution to maximize efficacy and avoid exaggerated weakness, while ensuring an individualized approach for each patient.
-which was in the 8 RCT the exact time of evaluation of the effect?
The time of evaluation of the effect was always at some point between Week 4 and Week 38 post injection. The exact timepoints for each study are available from the 'Results' columns of Tables 1 & 3.
-Did the authors use any guide during injections?
Comments about injection guiding techniques have now been added:
-in the new Tables 2 & 4
-in the lines 345-361:
- Injection techniques: As each has its advantages and limitations, a combination of the following BT injection techniques may be an ideal approach:
--electromyography-guided injection makes it possible to hear the bursts of muscle contraction when the tremor occurs;
--electrical stimulation is precious as this is the sole technique ensuring that the functional effect of the stimulated muscle indeed corresponds to the tremor movements; however, it can be difficult to distinguish between contractions due to stimulation and those due to the tremor;
--ultrasound helps ensuring that the needle is in the targeted muscle and sometimes makes it possible to see the contraction during the tremor, but it does not make it possible to know if the BT is injected to a muscle area actually causing the tremor movements, which could mitigate its effectiveness.
Using anatomical landmarks is the least effective strategy, as it has been shown to be over-all less effective than the above-mentioned techniques [65]. Finally and importantly, the success of each of these techniques is highly dependent on the experience and skill of the clinician using it, therefore all the published comparisons have been biased by the respective skills of the individual investigators in mastering each of the techniques [65-67].
-did you find any common criteria in the selection of the muscles and the injection technique?
The muscle selection was either based on the clinical expertise of the physician or on kinematic or EMG assessments, or both. However, we could not find data in the literature relating criteria for muscle selection and criteria for the choice of the injection technique (electrical stimulation, ultrasound or EMG).
-How many patients were pre-evaluated (polygraphy, sensors, kinematics..) to analyze agonist/ antagonist muscle patterns before injections?
This is mentioned in the ‘Muscle selection’ column: “clinical judgement; EMG; kinematics...” of Tables 1 & 3.
We have however, added a sentence in that respect, lines 152-154:
Of note, a minority (n=30) of the enrolled participants were pre-evaluated using kinematics to analyse agonist/antagonist muscle patterns before injections [25].
-Which are the medium/maximum dosages used in the studies for the different toxins?
This data is displayed in the ‘Dosage & Dilution’ column (when provided in the studies) of Tables 1 and 3.
-How many patients received only one injection and how many had a follow-up with repeated injections and for how long?
This is also stated in the 'Dosage & Dilution' column of Tables 1 and 3, where we provide information about repeated injections whenever appropriate:
-Table 1, Jankovic 1996: +/- repeated with 100 U (if no efficacy in week 4)
-Table 3, Samotus 2016: injections at W0, W16 and W32
-Table 3, Samotus 2018: injections every 16 weeks and follow during 96 weeks
-Table 3, Samotus 2019: 3 bilateral injections during 30 weeks
-Table 3, Pullman 1996: "booster injections" every 10-14 days during first month
-Table 3, Samotus 2017: injections every 16 weeks and follow during 96 weeks
In all other studies, it is implied that there is only one injection.
-Did Herz's measurement of tremor was performed in some studies?
To our understanding, Herz’ electrodynamic measurement of tremor (Herz H, Nusselt L. Ein einfaches Gerät zur Tremorregistrierung [A simple method for measurement of tremor (author's transl)]. Biomed Tech (Berl). 1973 Dec;18(6):240-1) was not used in the studies we have reviewed.
COMMENT 7: I suggest adding a new table of mostly injected muscles.
All these results could definitively add value to the discussion.
We have added a column in that respect in Tables 2 & 4.
COMMENT 8: Line 147 please be clear about which paper reports (add REF) that toxin is effective up to 24 weeks, this is not typical clinical practice in BoNTA clinic.
We have added the following lines (179-181):
Of the eight RCTs, three were cross-over studies with wash-out periods of 12 to 16 weeks that could be viewed as short, as BT – this has been particularly examined for abobotulinumtoxinA - may retain efficacy beyond 24 weeks [30,31]
New references cited here are:
30) Esquenazi A, Delgado MR, Hauser RA, Picaut P, Foster K, Lysandropoulos A, Gracies JM. Duration of Symptom Relief Between Injections for AbobotulinumtoxinA (Dysport®) in Spastic Paresis and Cervical Dystonia: Comparison of Evidence From Clinical Studies. Front Neurol. 2020 Sep 25;11:576117. doi: 10.3389/fneur.2020.576117
> BT effective until week 28 in a significant proportion of patients.
31) Ojardias E, Ollier E, Lafaie L, Celarier T, Giraux P, Bertoletti L. Time course response after single injection of botulinum toxin to treat spasticity after stroke: Systematic review with pharmacodynamic model-based meta-analysis. Ann Phys Rehabil Med. 2022 May;65(3):101579.
> half-life of BoNT effect disappearance for A/Abo is 13.1 weeks
COMMENT 9: When discussing the results of the systematic review I would stress the lack of a standardized and objective method of weakness assessment and that the studies selected used different assessing scales for primary outcomes.
Thank you for this comment, we fully agree and have modified the sentence about methodological limitations at the end of the introductory paragraph of Discussion (lines 244-249):
Methodological limitations include low sample sizes, lack of standardized methods for assessing efficacy and for measuring weakness precluding rigorous comparisons between toxins, doses and dilutions, lack of multicentric protocols (only in one study), few intention-to-treat analyses, often with patients being excluded from the analyses because of injections deemed "ineffective" or because there was "exaggerated weakness".
COMMENT 10: How many studies consider the patient's point of view (pGIC)? is this relevant in the auto-detection of tremor in daily life activities? Probably this point has to be also stressed in the discussion.
Only few studies have considered pGIC as specified in the Tables 1 & 3: see the ‘Results’ column, where ‘patient's global impression of change’ appears. This may be highly impacted by self-detection of tremor in daily life activities. We have now added a point on this in Discussion, lines 249-252:
In addition, only few studies have directly evaluated the overall patient’s point of view, as in the Patient Global Impression of Change (PGIC) [24,25,28,29]. This is unfortunate as self-detection of tremor in daily life activities may be more readily captured by such a tool than by other assessment methods.
COMMENT 11: I consider too long the description of the pathogenesis of different tremor syndromes, even if interesting, in particularly for the cerebellar tremor. The mechanism of tremor is important considering the possible effect of BoNTA but has to be synthesized; the mechanism of dystonic tremor is also to be taken into account underlying why it could be more beneficial of BoNT treatment in comparison with other types of tremor (ie cerebellar tremor due to lesions or neurodegeneration).
Pathogenesis of the various tremor types is sometimes unclear to toxin injectors and we believe that it was important to summarize things in a way as didactic as possible in such a paper. In view of the Reviewer’s comment though, we have tightened up the paragraph on the pathophysiology of cerebellar tremor (lines 267-279).
On another front, the journal has asked us to write a rather long review of over 3000 words. We kindly ask you to consider this aspect as well.
The Reviewer is correct about dystonic tremor and we have added a whole paragraph to the discussion on this topic, lines 297-308:
4.3. Botulinum toxin in dystonic tremors
The pathophysiology of dystonic tremor remains unclear. Since the new syndromic understanding of the term dystonia promoted by Marsden and Fahn in the late 1970s [60,61], the scientific community has struggled to come up with a clear and universally accepted definition of dystonia [62,63]. In that context, it would appear that up to 50% of patients with dystonia develop some form of tremor, a condition that has been termed dystonic tremor [64]. In terms of therapeutic approach, and given the established efficacy of BT injections in focal dystonia and in the treatment of other tremor types, BT for dys-tonic tremor has received surprisingly little attention from the scientific community, with only one published placebo-controlled trial referred to in this review (Tables 1 & 2) [29]. The effectiveness of BT injections was particularly marked in that trial, however, it re-mains difficult to compare it with BT effectiveness in other tremor types.
COMMENT 12: The “Going further” paragraph lists and summarizes advice from the authors, so I would underline that these are opinions, possibly expert opinions, but they are not properly “lessons learned from this review” (line 255). The results of this review process needed to be underlined to be useful for the scientific community.
We have deleted the words ‘lessons learned from this review’ and have added a sub-paragraph heading ‘5.1. Advice based on personal experience’ to emphasize that these are personal opinions only.
COMMENT 13: Till now there is a lack of evidence of the efficacy of the rehabilitation approach in tremors. I will suggest reformulating and shortening this paragraph and clarifying the differences in the possible retraining techniques in different typologies of tremor. Anyway, this is not the main topic of this review.
+COMMENT 14: Regarding motor strengthening programs (line 300) it should be noted that not all tremor types would benefit from this kind of rehabilitation, ie dystonic tremor.
This is indeed not the central subject of the review and we have tightened up this paragraph as recommended by the Reviewer. However, to be useful to the reader, we have retained this part of the manuscript, as there is in fact controlled evidence on rehabilitation techniques and tremors, albeit limited. As mentioned earlier in the manuscript, the neurorehabilitative aspect of patient management is unfortunately still poorly understood by a large proportion of neurologists. In view of the early controlled evidence available (references 68, 71, 75 and 77 in particular), we believe that neurorehabilitation should be offered as an adjunct to botulinum toxin injections, which act as a symptomatic treatment for tremor for a few months, whereas daily strengthening or target reaching training could act as a longer-term treatment. We have also added a heading to the paragraph to clarify our thinking: '5.3. Neurorehabilitation techniques as an adjunct to botulinum toxin injections' (line 374).
We have also amended the concluding sentence: “Taking all these aspects into account, there is still much to be done to improve knowledge on botulinum toxin injections…” (deleting “in combination with neurorehabilitation for tremor”) (lines 395-396).
The Reviewer is also correct that rehabilitation techniques do not apply to all types of tremor, so we have clarified this: “Accuracy training has been shown to be effective for cerebellar disorders, including essential tremor, and is encouraged by physiotherapists” (lines 386-387) “Motor strengthening programs have also produced encouraging results in essential tremor and certainly deserve to be investigated…” (lines 389-390).
COMMENT 15: Citations are not completely adequate, papers on spasticity are cited several times with no particular congruity; furthermore, there is lacking evidence in the literature and in clinical daily life that tremor has to be treated with similar strategies.
+COMMENT 16: Cit 68-69-70-71 is not fully related to this subject.
The indicated references have been deleted as well as the sentences that referred to them.
COMMENT 17: Lastly, I would strongly highlight for the future the need for an upper limb tremor uniformed botulinum injection protocol, to start RCTs without too strong methodology issues.
We fully agree and have modified the last sentence of the article (lines 395-400):
Taking all these aspects into account, much still needs to be done to improve knowledge about BT injections for tremor. There is a need for some form of upper limb tremor uniform botulinum injection protocol usable for high-quality RCTs, with the objective to improve muscle targeting and the selection of dose and dilution to maximize efficacy and avoid exaggerated weakness, while ensuring an individualized approach for each patient.
Reviewer 4 Report
Comments and Suggestions for Authors
The article provides a comprehensive overview of the treatment of upper limb tremor, with particular emphasis on the BoNT injections. The analysis of the manuscript raises the following criticisms:
1. In the “Method” section, it is suggested to clearly state the inclusion and exclusion criteria;
2. In the “Discussion” section, most of the content of this part is about the pathophysiological mechanism of the disease, and it is suggested to comprehensively describe the clinical efficacy and safety of BoNT in the treatment of tremor;
3. The abbreviation and complete spelling of some words are not consistent in this paper, which makes the content look confused;
4. There are some grammar or punctuation errors to avoid;
5. The content of the table has duplicate header;
6. In the section " Going further", the content describes both the suggestions of botulinum toxin injections on tremor and rehabilitation physiotherapy for tremor, with slightly confused. It is suggested to describe different titles separately.
Comments on the Quality of English LanguageModerate editing of English language required
Author Response
The article provides a comprehensive overview of the treatment of upper limb tremor, with particular emphasis on the BoNT injections. The analysis of the manuscript raises the following criticisms:
- In the “Method” section, it is suggested to clearly state the inclusion and exclusion criteria;
Thank you for the comments. We have restructured the Methodology section using the following subsections:
2.1. Research Methodology and Inclusion Criteria
2.2. Screening Methodology and Exclusion Criteria
2.3. Data Collection
2.4. Quality Assessment Method
We have clarified one point in section 2.1 to explain that all articles that met the inclusion criteria already described were retained (lines 83-84): "Article type was filtered to retain all "Clinical Trials" and "Randomized Clinical Trials" (RCT)".
- In the “Discussion” section, most of the content of this part is about the pathophysiological mechanism of the disease, and it is suggested to comprehensively describe the clinical efficacy and safety of BoNT in the treatment of tremor;
We would like to respectfully respond to this observation with a few comments from our team:
- As stated several times in our article, the literature on botulinum toxin injections for tremor is sparse.
- Its efficacy must necessarily be referred to the pathophysiology of each type of tremor. In our opinion, it is not possible to discuss its efficacy on 'tremor' as a whole without insight on why botulinum toxin may or may not work on each type of tremor. Tremor is a symptom that can be seen in a wide range of conditions with various causes. With this in mind, we have elected to discuss its efficacy for each cause and have named our paragraphs accordingly (each section of the discussion is entitled “Botulinum toxin in XX”).
-We believe that the safety of its use is also adequately discussed: the tables show the adverse events noted in each study, virtually none other than exaggerated limb weakness. Weakness is a major limiting factor in the use of BT and is mentioned several times in the manuscript and also in the abstract.
- The abbreviation and complete spelling of some words are not consistent in this paper, which makes the content look confused;
+ There are some grammar or punctuation errors to avoid;
We have have corrected some spelling errors (some are highlighted). Please do not hesitate to let us know of any other errors you may find, pending proofreading by the editorial team.
- The content of the table has duplicate header;
We assume that you are referring to the 'ref/n/BoNT/Muscle selection...' line in the tables and have removed the duplicates to keep it to a single line.
- In the section " Going further", the content describes both the suggestions of botulinum toxin injections on tremor and rehabilitation physiotherapy for tremor, with slightly confused. It is suggested to describe different titles separately.
We have re-stuctured this paragraph by adding the following sub-headings:
5.1. Advice based on personal experience
5.2. An 'asymmetric' injection pattern?
5.3. Neurorehabilitation techniques as an adjunct to botulinum toxin injections
Round 2
Reviewer 3 Report
Comments and Suggestions for Authors
Author Response
Many of the changes done to the review article are satisfactory but I ask the authors to work more on some details of the paper to make it acceptable
The explanation of dystonic tremor has been added but is not completely clear yet.
Tables are now clearer and more information has been added.
RESPONSE: Please find our responses below. All changes in the document are tracked using green highlighting. We have now tried to simplify and clarify the dystonic tremor paragraph.
COMMENT 1-Line 9: After “promising avenue”- I would add -“but there is still no evidence of efficacy.
RESPONSE: Since there has been repeated double blind placebo-controlled evidence of efficacy on tremor rating scales and on accelerometry, we have added the slightly modified phrase ‘but there is still no evidence of efficacy on upper limb function’ (lines 9-10).
COMMENT 2-Line 19: Conclusion: I would say: evidence is still lacking but some data have been analyzed regarding the modality and the site of injection (distal versus proximal) …… to be discussed….
RESPONSE: We added ‘Controlled evidence is still lacking but it is suggested that’, lines 19-20.
As argued in the body of the manuscript, these suggestions are based on early literature and on the long-term experience of the authors in practicing shoulder injections and ‘asymmetric’ distal injections avoiding or minimizing block on extensors and supinators.
COMMENT 3-Line 40: isometric: please explain the concept which is not clear.
RESPONSE: We have simply removed the specifications of kinetic, postural and isometric tremors in this sentence, which did not add much at this introductory stage of the manuscript.
COMMENT 4-Line 43: functional: please explain what do you mean: psychological or disabling
RESPONSE: Lines 41-48: We have slightly modified this paragraph to improve clarity. We have distinguished disabling "functional disturbances" (specifying the disrupted activities of daily life) and "psychosocial hardships" (where we describe the psychological and social consequences encountered by patients).
COMMENT 5-Line 99: please report clinical scales used (I would suggest to include scales in table 1 in a new column).
RESPONSE: We have made the following changes: lines 96-97 “…all reported outcomes including the clinical scales used…”.
In fact the scales used are already listed in the Results column of Tables 1 and 3 ("FTM, NIGCGC, PGIC, Bain, etc."). We believe that listing them in a whole new column may prove too cumbersome for the tables.
COMMENT 6-Line 202: kinematic analysis: please explain if you mean optoelecronic system, accelerometry gyroscopy?
RESPONSE: We have added the following specifications: lines 200-205 "Interestingly, the published kinematic analyses used motion sensor devices including three goniometers and one torsiometer placed over forearm, wrist, elbow and shoulder joints during two postural tasks and two weight-bearing tasks; at the shoulder level, these kinematic analyses assessed shoulder flexion-extension and abduction-adduction but it is not clear whether they also considered shoulder rotation [34-36]."
COMMENT 7-It would be interesting to create a column in Table 1 focused on quantitative analysis: which type and if it has been used or not used
RESPONSE: It would indeed be interesting to distinguish in Table 1 the clinical from the quantitative analyses. However, we have assessed that this might render the massive table illegible in terms of script size; we have thus opted for a compromise which is the current version in which clinical and quantitative data are superimposed in the same Results columns.
As for the ‘Muscle Selection’ column, we also consider that an additional column may be redundant, as when they are used, kinematics are already specified in that column of Tables 1 and 3 ("clinical judgement", "kinematics", "EMG assessment", "kinematics with automated algorithm", etc.).
COMMENT 8-Is there a role of kinematic analysis in future studies to test the patient during specific tasks
RESPONSE: Absolutely. We have pooled the response to this comment with the Response to COMMENT 13.
COMMENT 9-Lines 298-308: please improve 4.3 paragraph.
RESPONSE: As stated above, we have now simplified the dystonic tremor paragraph to try and bring more clarity.
COMMENT 10-Lines 335: about dilution: 300-400 U U/ml for abobotulinum: would you also consider 500 U as it is in may countries?
RESPONSE: Yes, we have made the change, line 336.
COMMENT 11-Please include a specific reference for every guide at the end of every paragraph (lines 348/352/356)
RESPONSE: References have been added (lines 349 – 353 – 357).
COMMENT 12-The “Going further” paragraph lists and summarizes opinions from the authors and has been clarified but I ask the authors to modify the title of 5.1: is too strong; I suggest to change “advice based on personal experience” in “Comments and suggestions for best practice”…
RESPONSE: Correction implemented (line 314).
COMMENT 13-Lines 395-400 This paragraph should be use to propose operative advices ato the scientific community in order to conduct new RCTs.
RESPONSE: We agree and have added the following paragraph (lines 401-406):
“Improving these aspects could involve the use of kinematics. By analyzing angular movements applied to each joint in different tasks, these tools may allow objective identification of the muscles to be targeted, as well as tremor monitoring after BT injection [34-36]. This could be supported by automated algorithms, albeit under investigator control. While visual assessment remains key and the experience of the examiner plays a major role, they may be enhanced by kinematics for diagnostic accuracy and objectivity.”
COMMENT 14-There are some statements that are not completely evidence based: we do not know if acting only on agonist muscles could be as effective as acting on both agonist and antagonist muscles. Maybe this could be true just for a subtype of tremor and must be underlined (line 370).
RESPONSE: We agree and have added a comment of restriction at the end of the paragraph (lines 374-375): “Obviously, this concept of ‘asymmetric’ injections needs to be demonstrated in well-designed clinical trials.”
COMMENT 15-The paragraph about rehabilitation techniques still lack of any reference about dystonic tremor.
RESPONSE: We found no studies pertaining to rehabilitation in dystonic tremor. We have therefore added the following sentence (lines 393-394): “Evidence for the rehabilitation of dystonic tremor is still lacking”.
COMMENT 16-Line 391: please be clear about which paper reports (add REF) that rehabilitation is the only solution in this case of tremor.
RESPONSE: There is no evidence in the literature to support such statement. We were only discussing that given the difficulties associated with toxin-induced weakness of the finger and wrist extensors (discussed earlier in the manuscript), rehabilitation might prove one of the only solutions. However, to avoid confusion, we have now deleted the sentence.